# Harmful Effects on the Hippocampal Morpho-Histology and on Learning and Memory in the Offspring of Rats with Streptozotocin-Induced Diabetes

**DOI:** 10.3390/ijms252111335

**Published:** 2024-10-22

**Authors:** Marcela Salazar-García, Laura Villavicencio-Guzmán, Cristina Revilla-Monsalve, Carlos César Patiño-Morales, Ricardo Jaime-Cruz, Tania Cristina Ramírez-Fuentes, Juan Carlos Corona

**Affiliations:** 1Laboratorio de Investigación en Biología del Desarrollo y Teratogénesis Experimental, Hospital Infantil de México Federico Gómez, Mexico City 06720, Mexico; villagu@yahoo.com (L.V.-G.); cpatino@cua.uam.mx (C.C.P.-M.); 2Unidad de Investigación en Enfermedades Metabólicas, Centro Médico Nacional Siglo XXI, Instituto Mexicano del Seguro Social, Mexico City 06725, Mexico; macrisrev@gmail.com; 3Departamento de Ciencias de la Salud, Universidad Tecnológica de México-UNITEC México-Campus Sur, Mexico City 09810, Mexico; ricardo.jaime.cruz@gmail.com; 4Sección de Estudios de Posgrado e Investigación, Escuela Superior de Medicina del Instituto Politécnico Nacional, Mexico City 11340, Mexico; tania.ramirez3190@outlook.com; 5Laboratory of Neurosciences, Hospital Infantil de México Federico Gómez, Mexico City 06720, Mexico

**Keywords:** cognitive dysfunction, hippocampus, hyperglycemia, learning and memory, morpho-histology, offspring

## Abstract

Learning alterations in the child population may be linked to gestational diabetes as a causal factor, though this remains an open and highly controversial question. In that sense, it has been reported that maternal hyperglycemia generates a threatening condition that affects hippocampal development in offspring. The pyramidal cells of the CA3 subfield, a key structure in learning and memory processes, are particularly important in cognitive deficiencies. We evaluate the effect of the hyperglycemic intrauterine environment on hippocampal histomorphometry in offspring, correlating it with spatial learning and memory, as well as the morphology of dendrites and spines in 30-day-old pups (P30). The maternal hyperglycemia affected the body weight, height, and brain size of fetuses at 21 days of gestation (F21), newborn pups (P0) and P30 pups from diabetic rats, which were smaller compared to the control group. Consequently, this resulted in a decrease in hippocampal size, lower neuronal density and cytoarchitectural disorganization in the CA3 region of the hippocampus in the offspring at the three ages studied. The behavioral tests performed showed a direct relationship between morpho-histological alterations and deficiencies in learning and memory, as well as alterations in the morphology of the dendrites and spines. Therefore, knowing the harmful effects caused by gestational diabetes can be of great help to establish therapeutic and educational strategies that can help to improve learning and memory in children.

## 1. Introduction

Learning difficulties are a significant cause of school failure in children and adolescents, leading to concentration issues and memory problems, which in turn affect their quality of life and disrupt their social, emotional, and behavioral development. Metabolic alterations during pregnancy, including gestational diabetes, are medical conditions that may contribute to cognitive problems in offspring [1,2]. There is evidence linking gestational diabetes to an increased risk of neurodevelopmental disorders, such as attention-deficit/hyperactivity disorder (ADHD) and autism spectrum disorder (ASD) in offspring [1,2,3]. However, the neurocognitive consequences of gestational diabetes remain a controversial topic. Currently, there is no consensus on whether maternal diabetes, either gestational or pre-gestational, is the primary factor affecting cognitive ability in offspring or if other factors, such as socioeconomic status, also play a significant role [4]. Clinical and experimental evidence using various scales has identified delays in neurological development, including learning disorders, anxiety, and hyperactivity, suggesting that gestational diabetes negatively impacts the health of the offspring [5,6,7,8]. Some studies have explored the association between gestational diabetes and cellular events in the hippocampus, a key structure involved in learning and memory with significant plasticity, making it a particularly vulnerable and sensitive region of the brain. However, further research is needed to fully understand the underlying mechanisms [9,10,11].

In humans, generating morpho-histological and morphometric evidence to understand the harmful effects of gestational diabetes and correlate these with cognitive and behavioral disabilities in children is not feasible. Consequently, animal models, such as rats, have proven to be invaluable tools for investigating these issues. Streptozotocin (STZ) is commonly used to induce gestational diabetes in rats in a controlled manner, replicating symptoms and effects similar to those observed in humans [12,13,14]. Rats share similar embryonic, anatomical, and physiological characteristics with humans, enabling the control of physical, nutritional, anthropological, and social factors [15]. As with humans, extensive data have associated diabetes in adult animal models with neurological complications and cognitive dysfunctions [16,17]. Alterations in several cellular processes, such as decreased neuron density and apoptosis in the hippocampus, have been used to evaluate and determine their potential involvement in the cognitive deficiencies observed in the offspring of diabetic rat mothers [18,19,20].

Previously, we demonstrated that STZ-induced gestational diabetes in rats on day 5 of pregnancy deregulated the intrauterine developmental timeline, restricted embryo-fetal growth, and delayed the maturation and remodeling of structures derived from neural crest cells [12]. However, there is currently limited evidence linking the harmful effects of a hyperglycemic intrauterine environment on brain histomorphometry, particularly in the hippocampus, with alterations in learning, spatial memory, and hippocampal neuronal plasticity compared to offspring from euglycemic rats. Thus, in this study, we conducted a morpho-histological analysis of the brain, focusing on the dorsal hippocampus and pyramidal neurons in the CA3 region in the offspring of diabetic and control rats at stages F21, P0, and P30. Finally, we performed behavioral tests to assess learning and memory.

## 2. Results

Gestational diabetes causes the fetus to develop in a hyperglycemic intrauterine environment, which can lead to postnatal cognitive deficits. Therefore, in this study, we used rats injected with STZ as a model of gestational diabetes and analyzed the hippocampus histomorphometrically in the offspring. This analysis was correlated with spatial learning memory and neural plasticity and compared with euglycemic offspring.

### 2.1. Maternal Body Weight and Metabolic Control in the Offspring of Diabetic Rats

The induction of diabetes in the pregnant rats adversely affected their health, as evidenced by reduced physical activity and the presence of polyphagia, polyuria and polydipsia. From day 18 of gestation until delivery, these rats did not gain body weight compared to the control (CN) group. A statistically significant difference in weight between the groups was observed from day 18 of gestation, with the CN pups having a gestation period of 21 days, whereas the diabetic rat (DR) group had an extended gestation period of approximately two additional days (Figure 1B). Before diabetes induction on day 3, blood glucose levels in the CN and DR groups were within normal euglycemic values (Figure 1C). However, following a single dose of STZ administered on the fifth day of gestation to the DR group, blood glucose levels increased dramatically, reaching values exceeding 200 mg/dL, which were maintained until the end of pregnancy (Figure 1C). Finally, only fetuses, neonates and pups exhibiting normal body flexion and no somatic malformations were included in the analysis. 

### 2.2. Changes in the Brain Morphometry at F21, P0 and P30 of Offspring of Diabetic Rats 

We further evaluated the effects of hyperglycemia on the brain morphometry in the offspring at F21, P0 and P30. At F21 and P0, the CN pups were euglycemic, while the DR pups were similar to those observed in the P30 CN group, as detailed in Table 1. The body size of CN pups at F21 was significantly larger compared to that of DR pups at F21. In contrast, no statistically significant difference was found in the body sizes of DR pups at P0 (born on day 23 of gestation) and in the CN pups at P0. On the other hand, by P30, the body size of CN pups was statistically larger than that of DR pups (Table 1). Additionally, the average brain weight across the different ages studied was statistically lower in the DR group compared to the CN group (Table 1). Finally, the measurement of the longitudinal and transverse axes of both cerebral hemispheres was also significantly smaller in the offspring of the DR group compared to those in the CN group (Table 1).

### 2.3. Alterations in Hippocampal Size and Neuronal Morphology in the CA3 Region in the Offspring of Diabetic Rats at F21, P0 and P30 

Later, to examine whether the hyperglycemia in the offspring alters the hippocampal size and neuronal morphology in the CA3 region at F21, P0 and P30, we analyzed the offspring of the DR group at the three stages. These results showed a statistically significant reduction in the dorsal hippocampal perimeter in the DR group compared to the offspring of the CN group (Table 2). It is important to emphasize that even though the P0 pups from both the CN and DR groups had the same body size at birth, the hippocampal size was smaller in the DR group.

In the CA3 region, the F21, P0 and P30 pups from the CN group showed 4 to 5 compact cell layers with more organized, pyramidal-shaped neurons (Figure 2A,C,E). 

In contrast, the DR group at all three ages showed more disorganized areas of pyramidal neurons in the CA3 region, with notable acellular spaces (Figure 2B,D,F). Specifically, at F21, the cells in the DR group did not present a typical pyramidal shape; instead, they were elongated with no well-defined limits (Figure 2B). The organization of the stratum pyramidalis and the neuronal morphology in the DR pups at P0 was similar to that observed in the CN pups at F21 (Figure 2A,D). At P30, the pyramidal layer of the DR pups showed less morphological cell maturation compared to the CN group (Figure 2E,F). Regarding neuronal density in the CA3 region at F21 and P30, it was higher in the CN group compared to the DR group. However, at P0, the difference in neuronal density between the two groups was not statistically significant (Table 2).

### 2.4. Impact of Gestational Diabetes on Locomotor Activity and Cognitive Functions in Rat Offspring

To evaluate whether the hyperglycemic intrauterine environment altered locomotor activity in the offspring of the DR group compared to the CN group at P30, we analyzed various movement parameters. The DR pups showed a statistically significant decrease in the total distance traveled compared to the CN pups (Figure 3A). This disparity was due to a statistically significant decrease in horizontal movement in the DR pups relative to the CN pups (Figure 3B). Furthermore, in terms of rearing behavior in the open field (measured by vertical movements), the CN pups showed more vertical activity, while the DR pups displayed a significant reduction (Figure 3B). Although hyperglycemia did not affect anxiety levels in the offspring, DR pups showed a tendency to stay less at the periphery of the open field, indicating a slight decrease in thigmotaxis movements, though this was not statistically significant when compared to the CN pups (Figure 3B). The NORT was conducted to determine whether a hyperglycemic intrauterine environment adversely affects memory functions in the offspring at P30. A significant difference was observed between both groups in total exploration time, with CN pups spending more time in continuous exploration compared with the DR group (Figure 3C). Moreover, when one of the familiar objects was removed and replaced with a novel object, the CN group spent significantly more time exploring it compared to the DR group (Figure 3C). These data suggest that gestational diabetes, under the present experimental conditions, delays or modifies cognition, as evidenced by the DR group spending relatively less time exploring and interacting with both familiar and novel objects.

Later, the learning and memory capabilities of both groups were assessed using the MWM test to investigate whether hyperglycemia-induced cognitive impairments in the offspring of diabetic rats at P30. The results indicated that the escape latency to locate the platform was prolonged in the DR group compared to the CN group. Notably, the DR pups exhibited a tendency to follow indeterminate swimming paths until the third day of testing, after which they demonstrated improved search behavior directed toward the quadrant containing the platform (Figure 4A). During the probe test, in which the platform was removed, the time spent in each quadrant was recorded, as shown in Figure 4B. The data revealed that the time spent in the target quadrant (quadrant 4) was significantly reduced in the DR group relative to the CN group (Figure 4B). Moreover, the DR pups spent an extended period in quadrant 2, indicating an inadequate acquisition of spatial memory. These findings suggest that gestational diabetes adversely affected memory, as evidenced by the MWM test results. 

### 2.5. Morphological Changes in Hippocampal Pyramidal Neuron Dendrites and Spines in the Offspring of Diabetic Rats

In the offspring obtained from both the CN and DR groups, subjected and not subjected to the MWM test, we analyzed the morphology of dendrites and spines in hippocampal pyramidal neurons to assess whether the hyperglycemic intrauterine environment induced morphological alterations by P42. Golgi–Cox staining was employed to visualize these structures. Representative images of pyramidal neurons in the CA3 region are presented in Figure 5. Initially, we quantified the number of primary processes originating from the neuronal soma, including the apical dendrite, as well as the number of primary and subsequent branches. In rats not subjected to the MWM test, the pyramidal neurons in the CA3 region of the hippocampus exhibited characteristic morphology in both groups, featuring a large soma shaped like a triangular pyramid, a single axon, a basal dendrite, and a prominent apical dendrite, with the longitudinal extension exceeding the transverse (Figure 5A–D).

Further analysis of dendritic morphology revealed that at P42, pyramidal neurons in the CN group displayed an apical dendrite with numerous branches and densely populated dendritic spines along the axis (Figure 5A,B). In contrast, the DR group exhibited a reduced number of branches, and in some instances, sections along the axis lacked dendritic spines (Figure 5C,D), suggesting impaired dendritic formation in the pyramidal neurons of the DR group. Moreover, in offspring from both groups that were subjected to the MWM test, a greater number of branches and dendritic spines were observed in the CN group (Figure 5E) compared to the DR group (Figure 5F).

In rats that were not subjected to the MWM test, the total density of dendritic spines in the first branch of the apical dendrite was significantly higher in the CN compared to the diabetic rat DR group (Table 3). To assess spine morphology, we classified the spines in the CN and DR groups at P42 into four categories: long, thin spines, mushroom spines, stubby spines, and other types. Long, thin spines were the most prevalent in the CN group and significantly more abundant than those in the DR group. Similarly, mushroom spines were more numerous in the CN group than in the DR group. The stubby spines were found in comparable numbers in both groups, while the “other” types of spines were significantly more common in the CN group compared to the DR group (Table 3). 

These results indicate that dendritic spine formation is impaired in the pyramidal neurons of the DR group. In relation to the pattern of spine morphology in the P42 offspring from both groups subjected to the MWM, the total density of dendritic spines, including the long thin type, mushroom type, stubby type, and other spine types, was comparable between the two groups. No statistically significant differences were observed in any spine type (Table 3).

Finally, we evaluated dendritic arbor complexity using Sholl analysis, which involved measuring the number of intersections of dendrites with concentric circles from the soma to the distal ends of the dendrites. Both the CN and DR groups exhibited an increase in the number of intersections within the first 20 µm, peaking at 50 µm from the soma. Beyond this distance, the number of intersections gradually decreased, with no further arborizations observed at 150 µm. However, no significant differences were observed in the plasticity of the basal dendrites between the two groups.

## 3. Discussion

In summary, the findings of this study indicate that gestational diabetes had a significant impact on body weight, height, brain weight, and hippocampal size and led to a notable reduction in pyramidal cell density within the CA3 region of the hippocampus in the offspring of diabetic rats. Additionally, elevated blood glucose levels were associated with alterations in dendritic structural morphology in the CA3 region, characterized by decreased dendritic spine density and reduced dendritic length in the DR group. These structural changes were correlated with behavioral impairments observed in the DR group. Consequently, these results may be relevant to understanding the cognitive impairments associated with diabetes mellitus in humans [21].

Our results demonstrate that gestational diabetes significantly affected body weight and height in the offspring of rats. This reduction in weight may be attributed to elevated blood glucose levels in the intrauterine environment during gestational diabetes, which can lead to neonatal macrosomia and neurodevelopmental delays, as previously reported [22]. Additionally, male offspring of DR were found to be more susceptible to weight disturbances compared to female offspring, a finding linked to elevated maternal blood glucose levels [23]. These observations are consistent with earlier reports showing reduced body weight in the offspring of diabetic rats [12]. Reduced weight during pregnancy may result from neural tube anomalies in the fetus or serve as an indicator of underlying factors contributing to congenital disabilities [24]. Developmental indicators are crucial for evaluating the maturation of neonatal neurological responses and predicting behavioral changes in adulthood [25]. In this context, proper development and function of the hippocampus are essential for cognitive processes, particularly in the recognition of memory and the conversion of short-term memory into long-term storage [26].

Gestational diabetes resulted in notable reductions in brain weight and hippocampal size in the offspring of rats, along with a significant decrease in the density of pyramidal cells in the CA3 region of the hippocampus. These findings align with and extend previous observations in the same animal model, where gestational diabetes led to neuronal loss in the CA1 and CA3 regions of the hippocampus in offspring at P7 and P21 [27]. Alterations in the hippocampus due to maternal hyperglycemia, such as developmental delays, apoptosis, and neuroinflammation, have also been reported [22]. Moreover, gestational diabetes has been shown to decrease total neuronal density in the hippocampus and induce neuronal cell apoptosis [20]. The offspring of diabetic rats exhibited reduced hippocampal size, including decreases in the volumes of the CA1 region, dentate gyrus, and subiculum, along with a reduced number of cells [28]. Recent studies have demonstrated that gestational diabetes induces oxidative stress in the cerebral cortex and hippocampus of rat offspring, characterized by increased lipid peroxidation, ROS, abnormalities in glutathione metabolism, and decreased antioxidant levels in the brain [23]. Additionally, this study showed that adult females from diabetic rat offspring exhibited reduced anxiety levels, as assessed using the elevated plus maze and OFT. Young females showed deficiencies in spatial learning, evaluated with the MWM and Radial Arm Maze, while adult males exhibited short-term memory impairments [23]. These findings are consistent with the cognitive impairments observed in the present study, where P30 pups from the DR group displayed significantly reduced total distance traveled in the OFT, delayed cognition in the NORT, and impaired spatial learning and memory in the MWM test.

Moreover, maternal diabetes led to increased exploration of open arms in the elevated plus maze, a heightened preference for novel objects in the NORT, and impaired behavioral flexibility, as well as increased excitability of hippocampal neurons and a proinflammatory state, indicated by decreased receptor expression for advanced glycation end products (RAGE) [29]. Gestational diabetes also resulted in decreased recognition memory (measured by the NORT), reduced anxiety levels and inattention (measured by the OFT), and decreased hippocampal synaptic integrity with increased inflammation, correlating with reduced density and abnormalities in the CA1 layer [30]. Additionally, male offspring of gestational diabetic mice demonstrated increased repetitive behaviors in the three-chambered social interaction test, with alterations in genes related to forebrain development and neurodevelopmental gene networks in the striatum [31].

Observations from Golgi staining in the present study revealed that in the CA3 region of the hippocampus, pyramidal neurons in the offspring of DR at P42 exhibited reduced dendritic arborization and fewer dendritic spines. These findings suggest that gestational diabetes impairs the proper formation of dendrites and spines in pyramidal neurons. This result is consistent with a prior study showing that gestational diabetes adversely affects primary motor cortex lamination and neuronal function in offspring, leading to alterations in cortical neuron migration, cell polarity, neurogenesis, and dendritic arborization, which in turn reduce the excitability of deep-layer cortical neurons [32].

Furthermore, research on adult rats with diabetes induced by STZ demonstrated a decrease in dendritic spine density and dendritic length in pyramidal cells across the hippocampus, prefrontal cortex, and occipital cortex, with the CA1 hippocampal region being particularly affected [33]. STZ-induced diabetes in 4-week-old rats resulted in reduced dendritic branches and spine density in the parietal cortex, but not in control rats. Additionally, the STZ-induced diabetic group displayed impaired performance in the MWM test [34].

Gestational diabetes has been shown to alter metabolites in fetal mouse brains and disrupt hippocampal DNA methylation and gene regulation associated with cognition, suggesting a potential mechanism for the negative neurocognitive effects of gestational diabetes in offspring [35]. In children exposed to gestational diabetes, brain magnetic resonance imaging revealed reduced thickness in a portion of the left inferior body of the hippocampus corresponding to the CA1 subfield. This exposure appears to affect the shape of the hippocampus, particularly the CA1 subfield in both boys and girls and may reduce the volume of the right hippocampal CA1 subfield in boys alone [36]. 

Gestational diabetes has been linked to impairments in hippocampal development, which may contribute to neurocognitive and neurobehavioral disorders such as ASD and ADHD in offspring. Synaptophysin, a marker for synaptic density and synaptogenesis, as well as a key player in learning and memory processes, has been shown to have decreased mRNA expression in hippocampal neurons of offspring from diabetic rats at P7 and P14 [9]. This suggests that gestational diabetes adversely affects synaptogenesis in the hippocampus of the offspring. On the other hand, maternal hyperglycemia has been associated with reduced expression of brain-derived neurotrophic factor (BDNF), neuroinflammation, apoptosis in the hippocampus, and cellular damage in the dentate gyrus, all of which could contribute to neurodevelopmental deficits [22]. BDNF is crucial for neuronal plasticity, differentiation, and synaptogenesis and also has cytoprotective effects on pancreatic β cells [37]. Although maternal diabetes did not alter hippocampal BDNF expression at P0, a marked downregulation of BDNF was observed in both hippocampal hemispheres of female and male offspring from the diabetic group at two weeks. Furthermore, the density of BDNF^+^ cells was significantly reduced in both the right and left dentate gyrus of the diabetic offspring across all postnatal days analyzed [38]. Maternal high fructose diet-induced metabolic dysfunction in the hippocampus in adult female rat offspring, since GPR43, a butyrate receptor was downregulated in the hippocampus also in primary astrocyte culture from female maternal HFD offspring the GPR43 and the mitochondrial biogenesis was significantly suppressed, which was reversed with butyrate [39]. In addition, this study showed that the protein expression of hippocampal PSD95 was significantly suppressed [39]. PSD95 is a postsynaptic protein present in the majority of excitatory synapses and modulates postsynaptic function and maturation. It has been demonstrated that decreases in dendritic spines could be corroborated by the biochemical assessment of PSD95.

## 4. Materials and Methods

### 4.1. Animals

Sprague-Dawley rats weighing between 250 and 300 g were used in all the experiments. Rats were obtained from the Animal Facilities of the National Medical Centre Siglo XXI, Mexican Institute of Social Security (IMSS, for its acronym in Spanish). Females and males were acclimatized for 2 weeks before mating in separate cages and were housed under controlled conditions of temperature (22 ± 2 °C), humidity (50% ± 10%) and a 12-h/12-h light/dark cycle. Animals had free access to food and water. All procedures adhered to the Mexican Official Standard (NOM-062-ZOO-1999 [40]) guidelines for the care and use of laboratory animals were approved by the ethical and research committees of our institution (HIM2018-046 and HIM2016-037).

### 4.2. Cycle Determination and Mating 

Subsequent to the acclimatization period, vaginal smears stained with 0.5% toluidine blue were used to determine the time at which females were sexually receptive. During the estrus stage, blood and urine glucose levels were measured in females. A group of rats with these characteristics was then placed in a cage with a male, where they remained for approximately 12 h (7 p.m. to 7 a.m.). The presence of vaginal mucus plugs, sperm in the vaginal smear, or both was indicative of pregnancy, and it was determined as day 0 of pregnancy. Finally, pregnant rats were placed in individual cages under controlled conditions as in the acclimatization period [12].

### 4.3. Diabetes Induction in Rats

Pregnant rats were randomly separated into two groups (n = 8 per group and stage). Diabetes was induced following a previously described protocol [12]. Briefly, on the 5th day of pregnancy (preimplantation period), a single dose of STZ 50 mg/kg (Sigma, St. Louis, MO, USA) was administered intraperitoneally. In the diabetic rat (DR) group, rats with blood glucose levels ≥ 200 mg/dL 48 h after STZ administration were included as the experimental group (80% of rats subjected to STZ develop diabetes). The second group was designated as the vehicle-treated control (CN) group. Body weight, blood glucose, and glycosuria were monitored from the beginning of pregnancy until sacrifice in both groups, with special attention given to recording the length of gestation. From both groups, the fetuses of 21 days of gestation (F21) were obtained via cesarean section, newborn pups (P0) were delivered spontaneously, and pups of 30 postnatal days (P30) were also used for all the experiments. Additionally, a group of pups P30 was used for the behavioral studies and the hippocampal neuronal plasticity analysis. In this study, were used indistinctly female and male of the offspring rats.

### 4.4. Somatometric and Morphometric Analysis

Using a digital Vernier Calliper (Starrett), the crown–rump length, naso–occipital length, tail length, and waist circumference at the umbilicus were measured. The offspring from the DR and CN groups at F21, P0 and P30 were morphologically evaluated. Offspring showing any external malformation were excluded from the weight and somatometry analysis (Salazar Garcia et al., 2015 [12]). The brains from each group were removed, weighed, measured and photographed using an Axiocam MRc digital microscope camera (Zeiss, Oberkochen, Germany). Images were captured with the Axio Vision AC program (Version 4.4, Zeiss), and the major and minor axes of each cerebral hemisphere were measured. The brains were fixed in 4% paraformaldehyde, dehydrated through a gradient of alcohols and subsequently embedded in VIP 4005 paraffin (Sakura Finetek, Torrance, CA, USA). 

### 4.5. Cresyl Violet Staining

Cresyl violet staining was employed to quantify the number of cells in the CA3 region of the hippocampus. Tissue sections, 5 µm thick, were prepared, and the hippocampi were stained with cresyl violet. The fixed tissue sections were first immersed in 70%, 90%, and 100% ethanol for 15 min each, followed by re-immersion in descending ethanol concentrations. The sections were then stained for 5 min in a filtered cresyl violet solution and subsequently washed in distilled water. After staining, the sections were dehydrated again in ethanol, placed in xylene for 10 min, and coverslipped. The perimeter of the dorsal hippocampus was measured, and the CA3 field was delineated to evaluate the density of pyramidal neurons (Py). The number of neurons in the CA3 region was counted using an optical microscope (Olympus BH2-RFCA 1.25X, Tokyo, Japan) equipped with a Nikon Coolpix E995 (Tokyo, Japan) camera and analyzed with Fiji-ImageJ software (v.1.54f, NIH, Bethesda, MD, USA).

### 4.6. Behavioural Tests

The offspring of both groups were subjected to a battery of behavioral tests, which included the Open Field Test (OFT), Novel Object Recognition Test (NORT) and Morris Water Maze (MWM), one trial per day for three consecutive days. Twenty-four hours before the tests, the pups were handled and habituated. After each test, the apparatuses were cleaned with a 75% ethanol solution to remove any trace of odors. The behavioral tests were conducted between 8:00 p.m. and 10:00 p.m. in a dimly red-lit room isolated from noise. All behavioral tests performed were recorded on video and analyzed using Fiji ImageJ software (NIH, Bethesda, MD, USA). The procedures for the behavioral tests are illustrated in the schematic diagram of the experimental procedures in Figure 1A.

### 4.7. Open Field Test (OFT) 

The OFT was used to assess locomotor activity in the offspring of both groups at P30 (n = 8 per group). The apparatus is composed of opaque Plexiglas (90 × 90 × 90 cm^3^). Each pup was placed in the center of the apparatus and allowed to move freely for 10 min. The total distance traveled in the OFT was recorded, and horizontal and vertical movements, as well as thigmotaxis, were evaluated [41,42].

### 4.8. Novel Object Recognition Test (NORT) 

In the offspring at P30 of both groups (n = 8/per group), NORT was conducted, which is a test to recognize memory in rodent animals. According to the observation, rats prefer exploring novel objects over familiar ones. NORT consists of three phases: habituation, familiarization, and the test phase. During habituation, pups were placed in the center of an empty box measuring (50 × 50 × 30 cm) and were allowed to explore for 5 min. On the following day, during the familiarization phase, each pup was placed in a box with two identical objects for 10 min, after which it was returned to its cage. The test phase occurred after a 60 min inter-trial pause, during which the pups explored the box in the presence of one novel object and one familiar object for 5 min. Exploration was determined as time spent with the nose touching or sniffing an object within approximately 1 cm. The time spent exploring the familiar and novel objects was recorded. The novel object preference percentage was calculated as the proportion of time spent exploring the novel object relative to the total time spent exploring both objects, and this percentage was used for the statistical analysis [41].

### 4.9. Morris Water Maze (MWM)

The MWM test was employed to assess hippocampus-dependent spatial long-term memory and learning in the offspring of both groups. The apparatus consisted of a circular tank (diameter  =  110 cm; height  =  75 cm) filled with water (22  ±  1 °C), rendered opaque by the addition of powdered milk. The tank was divided into four quadrants: Target, Opposite, Sector 1, and Sector 2, each with a spatial cue on the tank wall. An escape platform (diameter  =  19 cm, height  =  22 cm) was positioned 1 cm below the water surface. The rats’ swimming activity was monitored via an overhead video camera and automatically recorded. 

The testing procedure lasted for 5 days. Initially, pups were trained over 4 consecutive days with four training trials per day, each beginning at the same time. During each trial, a pup was placed in one of the quadrants and allowed 1 min to search for the hidden platform. If the platform was not located within this time frame, the experimenter guided the pup to it. Once the platform was found, the pup remained on it for 15 s to memorize the spatial cues. Subsequently, the pup was placed in a cage for 15 s to rest before the next trial. The platform’s location remained constant throughout the experiment. On the final probe trial, the platform was removed, and the pup was placed in the opposite quadrant. The time spent in each quadrant was measured over 1 min. 

Learning ability was evaluated by recording the average latency to locate the platform (in seconds) across trials and the total number of successful attempts [43]. In the experiment, n = 16 pups were used and subdivided into two groups (offspring of P42 from the CN and DR groups subjected and not subjected to MWM). Data were recorded and processed using the Fiji-ImageJ software (NIH, Bethesda, MD, USA).

### 4.10. Golgi–Cox Staining

Golgi staining has been extensively used to determine dendritic morphology and to examine the morphology of hippocampal CA3 pyramidal neurons in both study groups (offspring subjected and not subjected to MWM); the brains of P42 pups were analyzed. Golgi–Cox staining was used to obtain hippocampal dendritic spine density via the Rapid Golgi Stain TM Kit (FD Neuro Technologies Inc) in accordance with the manufacturer’s instructions. Coronal tissue sections of 120 µm thick were cut at room temperature using a cryostat (Olympus BH2-RFCA). Following staining, the morphology of pyramidal neurons was examined. 

The dorsal hippocampus was identified, and six continuous slices were selected using an optical microscope (Olympus BH2-RFCA). Pyramidal neurons in the CA3 region of the hippocampus were identified by their characteristic triangular soma shape, apical dendritic extension toward the pial surface and abundant dendritic spines. Neurons were selected and photographed, tracking the path of their ramifications and the major and minor axis of the soma. Using an SPlan 100× oil objective with a numerical aperture of 1.25, different optical planes within the first 25 µm of the apical dendrite were examined. The long, thin, mushroom, stubby and other types of dendritic spines were identified and counted based on the classification criteria established by Sorra and Harris [44]. In the basal dendrite, a Sholl analysis was performed [45,46]. Concentric circles were drawn every 10 µm from the soma and up to 200 µm over the dendritic image. The number of dendritic branches intercepted by each circle was quantified to assess arborization and dendritic length. Two people, blinded to experimental groups, made the measurements and the values obtained were averaged. Data on spine densities and dendritic length were analyzed using one-way ANOVA, followed by the Bonferroni test for post hoc comparison.

### 4.11. Statistical Analysis

Statistical analysis was performed using GraphPad Prism Software (Version 8.01, GraphPad, Inc., La Jolla, CA, USA). Data are expressed as the mean ± Standard Error of the Mean (SEM) from at least eight independent experiments. To compare weight, height, maternal metabolic parameters, morphometric parameters, and behavioral tests of the offspring from CN and DR groups, we employed Student’s *t*-tests. Furthermore, statistical comparisons were made by analysis of variance (one-way ANOVA), followed by a post hoc Bonferroni test for group comparisons in behavioral tests. Differences were considered statistically significant when *p* < 0.05.

## 5. Conclusions

This research highlights that gestational diabetes induces neurobehavioral impairments and damage to hippocampal pyramidal neurons in the offspring of rats, with these effects persisting into the postnatal period. Further research is necessary to elucidate the underlying mechanisms driving hippocampal alterations caused by gestational diabetes in offspring. Given that gestational diabetes creates a hyperglycemic environment during fetal development, it may contribute to neurocognitive and neurobehavioral issues, damage to neuronal cells, and disruption of oxidative and antioxidative balance, leading to ROS accumulation, DNA damage, and inflammation. Therefore, well-designed research and clinical trials are crucial to confirm these detrimental hippocampal alterations induced by gestational diabetes and to develop educational and therapeutic strategies aimed at improving learning and memory deficits in affected offspring.

## Figures and Tables

**Figure 1 ijms-25-11335-f001:**
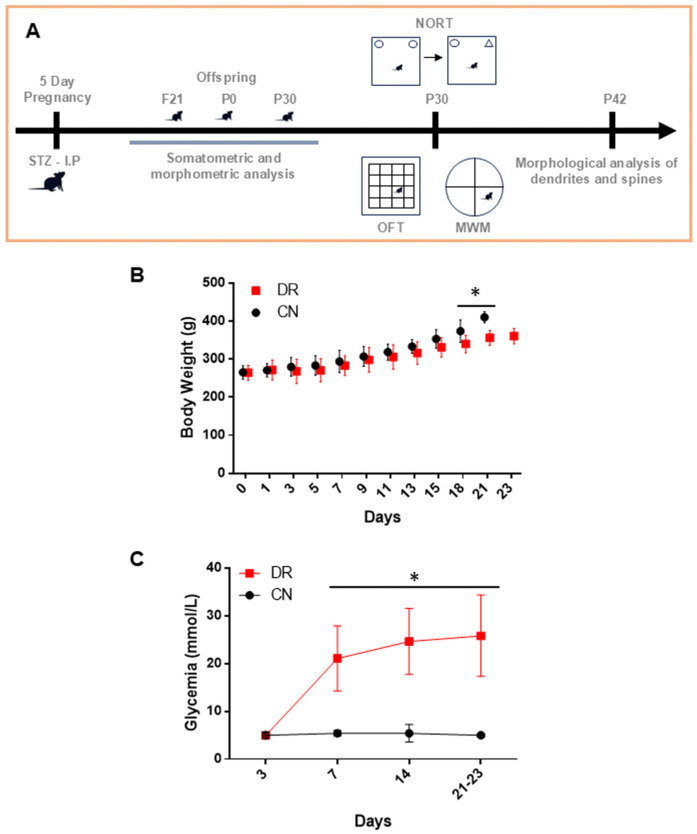
Maternal body weight and glycemic levels. (**A**) Schematic representation of experimental procedures and timeline. (**B**) Body weight (in grams—g) during gestation, with less weight gain observed in the diabetic mother group compared to the control group. (**C**) Blood glucose levels (in millimoles per liter—mmol/L) were measured on specified days, showing an increase in diabetic rats, with levels exceeding 200 mg/dL and maintained until the end of pregnancy. All results are presented as mean ± SEM of n = 8 rats per group. Significant differences between the CN group and DR group are indicated by * *p* < 0.05, as determined by a two-tailed unpaired Student’s *t*-test.

**Figure 2 ijms-25-11335-f002:**
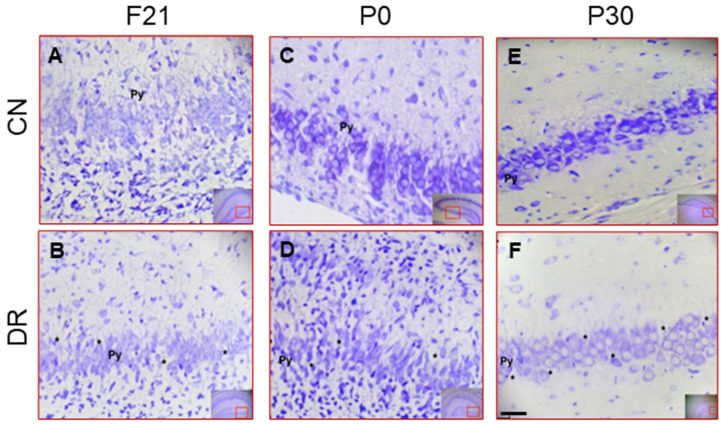
Decreased neuronal morphology in CA3 region in the offspring of diabetic rats. Representative photomicrographs of the CA3 region in the dorsal hippocampus are shown for fetuses at F21, P0, and P30 from the CN group (**A**,**C**,**E**) and the DR group (**B**,**D**,**F**) at the corresponding ages. In the DR group offspring, areas of disorganization are evident in the pyramidal cell layer (Py), with undefined boundaries and some acellular spaces (*) observed. The organization of the stratum pyramidale in P0 offspring from the DR group resembles that of F21 from the CN group. Additionally, a greater number of cells in the pyramidal layer (**B**) is evident in P0 and P30 offspring from the CN group and in F21 from the DR group offspring. The scale bar represents 500 µm. The red boxes (4× magnification) indicate the areas of higher magnification (20×) in each photomicrograph within the hippocampal region.

**Figure 3 ijms-25-11335-f003:**
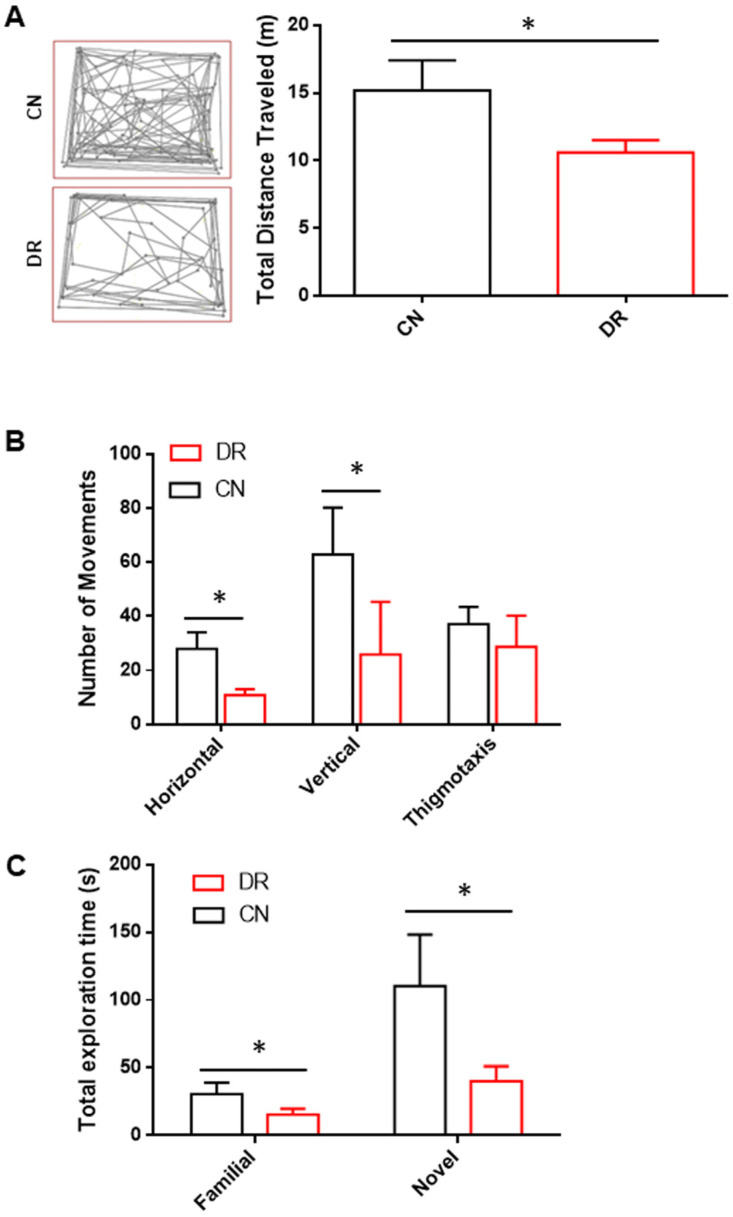
Decreased locomotor activity and altered memory function in offspring of diabetic rats. (**A**) Representative activity traces of the CN and DR groups and the total distance traveled by offspring from both the CN and DR groups were assessed using the OFT. Offspring in the DR group exhibited reduced locomotor activity compared to the CN group. (**B**) The total number of movements, including horizontal, vertical, and thigmotaxis movements, was measured in the OFT. The DR group showed decreased horizontal and vertical activity relative to the CN group. (**C**) In the NORT, the total exploration time for objects revealed a significant difference between the groups. The CN group spent more time exploring the objects than the DR group. When a familiar object was replaced with a novel one, the DR group spent less time exploring the novel object compared to the CN group. Results are presented as mean ± SEM of n = 8 rats per group. Significant differences between the CN and DR groups are indicated by * *p* < 0.05, as determined by a two-tailed unpaired Student’s *t*-test or by one-way ANOVA with post hoc Bonferroni tests.

**Figure 4 ijms-25-11335-f004:**
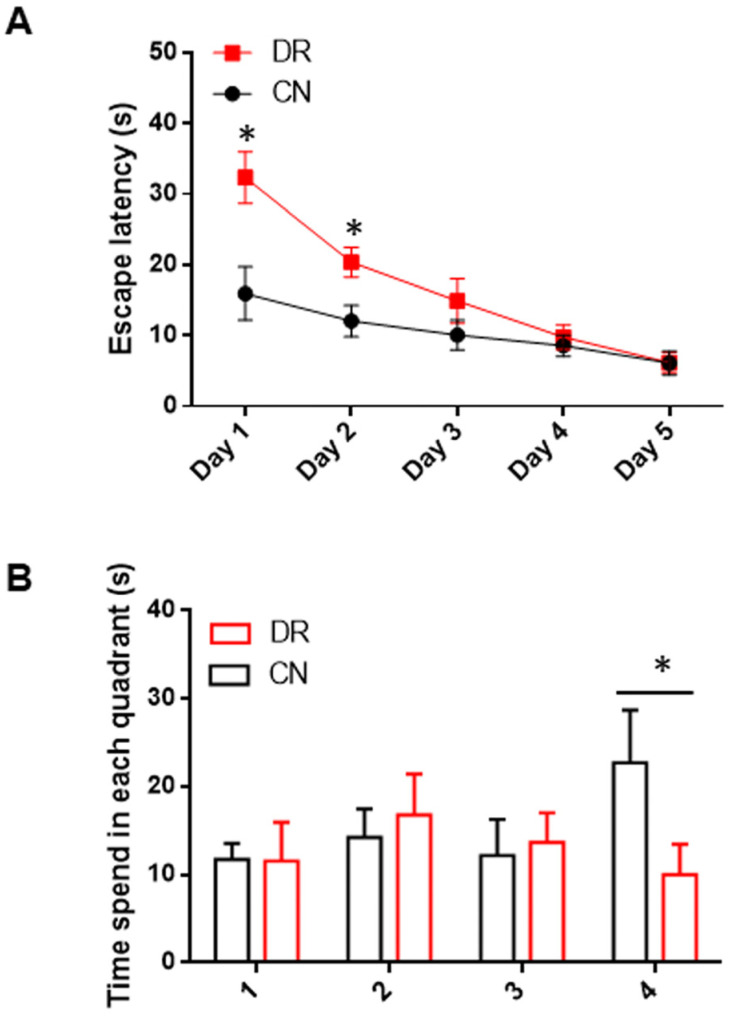
The hyperglycemic intrauterine environment decreased learning and memory ability in the offspring of diabetic rats. (**A**) The escape latency in offspring from both the CN and DR groups, measured over a period of five days or sessions in the MWM test, revealed that the DR group did not acquire the spatial task until the third day. (**B**) Analysis of the time spent exploring each quadrant in the MWM demonstrated that the DR group failed to learn the spatial task adequately. Specifically, this group spent less time in the target quadrant (Quadrant 4), where the platform was located, and exhibited dispersed attention across other quadrants. Results are presented as mean ± SEM of n = 8 rats per group. Significant differences between the CN group and DR group are indicated by * *p* < 0.05 by two-tailed unpaired Student’s *t*-test or one-way ANOVA with post hoc Bonferroni tests.

**Figure 5 ijms-25-11335-f005:**
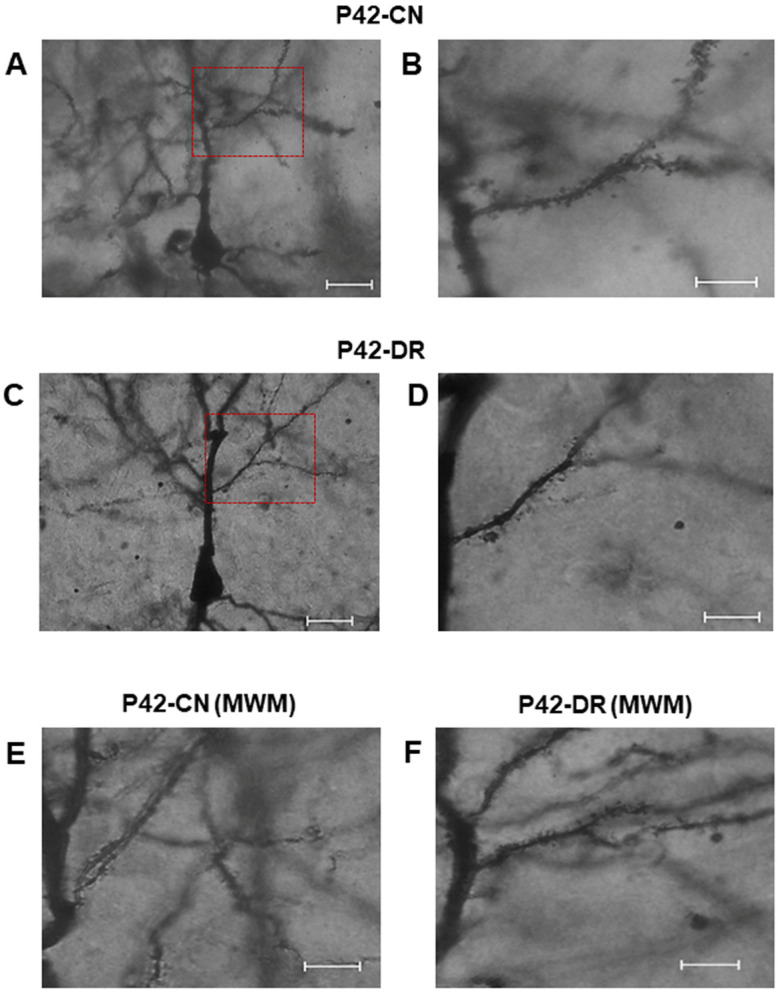
Decreased dendritic arborization and spines of hippocampal pyramidal neurons in the offspring of diabetic rats. Representative photomicrographs illustrate the apical dendrite and first branch of CA3 pyramidal neurons at P42 in offspring from the CN group (**A**,**B**) and the DR group (**C**,**D**). The DR group exhibited reduced dendritic arborization and a lower density of dendritic spines in the first branch compared to the CN group, indicating diminished synaptic efficiency. Following MWM testing, a greater number of dendritic branches and spines were observed in the CN group compared to the DR group (**E**,**F**). Scale bars are 25 µm (**A**,**C**) and 10 µm (**B**,**D**,**E**,**F**). The red boxes in panels (**A**,**C**) denote areas of magnification.

**Table 1 ijms-25-11335-t001:** Glycemia, somatometry and brain morphometry in the offspring of diabetic rats.

	F21-CN	F21-DR	P0-CN	P0-DR	P30-CN	P30-DR
Glycemia (mg/dL)	105 ± 3.08	262 ± 12.18 *****	79 ± 5.32	206 ± 11.59 *****	102 ± 1.4	112 ± 2.16
Body size (cm)	4.27 ± 0.29	3.72 ± 0.36 *****	4.46 ± 0.23	4.54 ± 0.27	14.20 ± 0.11	12.51 ± 0.29 *****
Brain weight (g)	0.32 ± 0.040	0.20 ± 0.06 *****	0.36 ± 0.05	0.26 ± 0.02 *****	1.83 ± 0.036	1.48 ± 0.105 *****
Longitudinal axis RH (mm)	7.74 ± 0.560	6.15 ± 0.73 *****	7.94 ± 0.59	6.52 ± 0.9 *****	14.47 ± 0.345	11.69 ± 1.54 *****
Transverse axis RH (mm)	4.57 ± 0.390	3.56 ± 0.56 *****	4.81 ± 0.36	4.07 ± 0.37 *****	8.06 ± 0.604	5.84 ± 0.485 *****
Longitudinal axis LH (mm)	7.72 ± 0.540	6.16 ± 0.7 *****	7.95 ± 0.58	6.66 ± 0.34 *****	14.21 ± 0.213	10.63 ± 0.87 *****
Transverse axis LH (mm)	4.12 ± 0.350	3.19 ± 0.49 *****	4.25 ± 0.31	4.12 ± 0.19 *****	7.87 ± 0.49	5.87 ± 0.550 *****

All measurements were conducted in fetuses at 21 days of gestation (F21), newborn pups (P0), and at 30 days postnatal (P30) from control rats (CN) and rats with diabetes-induced (DR). The data for all three developmental stages are presented as mean ± SEM with n = 8 rats per group. Statistical comparisons were made using Student’s *t*-test with significance set at * *p* < 0.05. Abbreviations used include right hemisphere (RH), left hemisphere (LH), centimeter (cm), millimeter (mm), gram (g), and milligram per deciliter (mg/dL). SEM values are indicated in parentheses.

**Table 2 ijms-25-11335-t002:** Hippocampal morphometry and neuronal density in the offspring of diabetic rats.

	F21-CN	F21-DR	P0-CN	P0-DR	P30-CN	P30-DR
Hippocampus (perimeter in mm)	1.42 ± 0.05	1.58 ± 0.04 *****	1.58 ± 0.13	1.07 ± 0.18 *****	2.34 ± 0.025	2.25 ± 0.025 *****
Neuronal density CA3 region (mm^2^)	87.64 ± 3.4	74.25 ± 2.15 *****	97 ± 12.6	95.44 ± 9.37	84.66 ± 15.76	32.75 ± 3.98 *****

All measurements were conducted in fetuses at F21, P0, and P30 from CN and DR groups. Data for these three developmental stages are presented as mean ± SEM with n = 8 rats per group. Statistical comparisons were performed using Student’s *t*-test, with significance denoted by * *p* < 0.05. SEM values are shown in parentheses.

**Table 3 ijms-25-11335-t003:** Density of dendritic spines of CA3 pyramidal neurons of the dorsal hippocampus in the offspring of diabetic rats.

	P42-CN	P42-DR	P42-CN (MWM)	P42-DR (MWM)
Total density dendritic spine (per 25 µm)	108 ± 8.05	75 ± 6.45 *****	146 ± 7.35	134 ± 6.05
Long thin (per 25 µm)	30 ± 8.24	23 ± 8.036 *****	56 ± 4.63	49 ± 5.12
Mushroom (per 25 µm)	22 ± 6.17	12 ± 4.02 *****	36 ± 5.37	35 ± 4.28
Stubby (per 25 µm)	17 ± 4.09	16 ± 3.96	27 ± 6.61	24 ± 3.11
Other (per 25 µm)	16 ± 5.51	7 ± 2.49 *****	27 ± 3.27	26 ± 4.78

All measurements were conducted on offspring at P42 from the CN and DR groups, including those not subjected to MWM. Additionally, the density of dendritic spines was assessed in offspring from both groups that underwent MWM. The data for the studied age are presented as mean ± SEM, with n = 8 rats per group. Comparisons were made using Student’s *t*-test with * *p* < 0.05. The SEM is indicated in parentheses.

## Data Availability

The original contributions presented in the study are included in the research article; any further inquiries can be directed to the corresponding authors.

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
