# Peer review of "Harmful Effects on the Hippocampal Morpho-Histology and on Learning and Memory in the Offspring of Rats with Streptozotocin-Induced Diabetes"

_ijms, 2024, doi:10.3390/ijms252111335_

Round 1
Reviewer 1 Report
Comments and Suggestions for Authors
The manuscript describes the impact of maternal STZ-induced hyperglycemia on offspring by orthodox neuroscience methods. Accordingly, certain hippocampus-related phenotypes are observed, but it is difficult to attribute these findings solely to hippocampal origins. For instance, motor initiation reluctance due to the weight gain may contribute to the poorer performance of DR mice. The authors then describe morphological changes, specifically decreases in spine density, in the hippocampal CA3. These changes correlate well with the observed phenotypes, but the molecular mechanisms that underlie these changes remain undescribed.
Minor
Table 1. Brain weight? (typo)
The authors have to describe the property of the 100x objective (the numerical aperture and the product name).
The authors are requested to quantify the thorny excrescences (if the slides are still available).
The gender of the offspring must be reported, and if relevant, separate statistics should be provided as supplemental material. As the authors described in the discussion section, maternal high-energy diet paradigms often reveal sexual dimorphisms in offspring phenotypes. The authors are requested to discuss how they treated the data regarding offspring sex.
Optional
Decreases of dendritic spines could be corroborated by the biochemical assessment of PSD95 (e.g., Wu et al. (10.1016/j.jnutbio.2024.109571)) or discussion/mentioning of that is ok.
Author Response
All changes to the manuscript text are shown in red.
Minor
Table 1. Brain weight? (typo)
Response: This is not a typo. We performed, among other things, the average body size and brain weight in the offspring of diabetic rats.
The authors have to describe the property of the 100x objective (the numerical aperture and the product name).
Response: We have now added the details in the methods section
The authors are requested to quantify the thorny excrescences (if the slides are still available).
Response: We thank the reviewer very much for this insightful comment. While the thorny excrescences are the postsynaptic components of synapses between mossy fibers of granule cells and dendrites of CA3 pyramidal neurons in the hippocampal formation.
There is not much quantitative data on the number and distribution of excrescences in rat offspring are available.
Because, first, clusters are of varying lengths and are distributed over hundreds of micrometers, making ultrastructural analysis excessively time-consuming.
Second, the vast majority are grouped into clusters and it is not possible to identify single excrescences within these clusters at the light microscope level.
Unfortunately, the slides are no longer available; however, it would be very interesting in the future to be able to carry out such an analysis.
The gender of the offspring must be reported, and if relevant, separate statistics should be provided as supplemental material. As the authors described in the discussion section, maternal high-energy diet paradigms often reveal sexual dimorphisms in offspring phenotypes. The authors are requested to discuss how they treated the data regarding offspring sex.
Response: We added a paragraph in the Methods: In this study, were used female and male of the offspring rats. Although we did not check if there was an effect on the diverse parameters performed between female and male rats, it would be interesting to analyze in the future if there are gender differences.
Optional
Decreases of dendritic spines could be corroborated by the biochemical assessment of PSD95 (e.g., Wu et al. (10.1016/j.jnutbio.2024.109571)) or discussion/mentioning of that is ok.
Response: We have now included some paragraphs in the discussion of the paper suggested by the reviewer and the relevance of PSD95 in dendritic spines.
Finally, We want to thank the reviewer for their valuable comments and constructive criticisms. We have taken this into account in this manuscript and for the future.

Reviewer 2 Report
Comments and Suggestions for Authors
The authors had investigated the harmful effects of gestational diabetes on the development of the hippocampus and neurons in rat offspring at various stages (F21, P0, and P30), examining how these changes affect learning and memory from both behavioral and tissue morphology perspectives. The topic was engaging and deserving of recommendation; however, there are several unclear points that need clarification and revision.
Q1. To establish the animal model, 8 pregnant rats were selected for each group and subjected to STZ injection. However, for the subsequent investigations, 8 rat offspring was used in each investigation. Did all the rats subjected to STZ develop diabetes? How were the rat offspring chosen for the following experiments individually?
Q2. Regarding Figure 2, an overview of the tissue morphology should be provided along with magnification details to enhance visibility of the specifics. Meanwhile, the scale bar should be unify presented for better clarity.
Q3. The movement traces of the rat offspring during the locomotor activity monitoring should be included in Figure 3.
Q4. Why was day 42 chosen to assess the development of hippocampal pyramidal neurons in rat offspring?
Q5. Why did the rats subjected to the MWM test show no significant differences in total dendritic spine density between the CN and DR groups?
Q6. What was the blood glucose level in the rat offspring? Was it recorded during the investigation?
Q7. Suggestions: The blood glucose levels in Figure 1 should be presented in mmol/L instead of mg/dL for greater clarity. Regarding Tables, the data should be presented as mean ± SEM rather than mean (SEM) for better clarity.
Comments on the Quality of English LanguageMinor editing of English language required.
Author Response
All changes to the manuscript text are shown in red.
Q1. To establish the animal model, 8 pregnant rats were selected for each group and subjected to STZ injection. However, for the subsequent investigations, 8 rat offspring was used in each investigation. Did all the rats subjected to STZ develop diabetes? How were the rat offspring chosen for the following experiments individually?
Response: We thank the reviewer very much for this insightful comment. On page 12 and in paragraphs 397-398, we indicate: In the diabetic rat (DR) group, rats with blood glucose levels ≥200mg/dL 48h after STZ administration were included as the experimental group (80% of rats subjected to STZ develop diabetes). On page 12 and in paragraphs 402-404, we indicate: From the DR group, the fetuses of 21 days of gestation (F21) were obtained via caesarean section, newborn pups (P0) were delivered spontaneously, and pups of 30 postnatal days (P30) were also used for all the experiments. In this study, we used indistinctly female and male of the offspring rats (page 12, paragraph 406).
Q2. Regarding Figure 2, an overview of the tissue morphology should be provided along with magnification details to enhance visibility of the specifics. Meanwhile, the scale bar should be unify presented for better clarity.
Response: We have now provided magnification details and unified the scale bar for better clarity.
Q3. The movement traces of the rat offspring during the locomotor activity monitoring should be included in Figure 3.
Response: We have now added traces in Figure 3.
Q4. Why was day 42 chosen to assess the development of hippocampal pyramidal neurons in rat offspring?
Response: After performing the MWM test (at P35) and allowing 7 days for rats to consolidate the acquired information, the acquired memory was evaluated again in the offspring obtained from CN and diabetic rats (at P42). Therefore, after that, we analyzed the morphological alterations by P42.
Q5. Why did the rats subject to the MWM test show no significant differences in total dendritic spine density between the CN and DR groups?
Response: Although there was no significant difference, there was a tendency for the total number of dendritic spines to decrease in the DR group. However, the DR group exhibited reduced dendritic arborization and a lower density of dendritic spines in the first branch compared to the CN group, indicating diminished synaptic efficiency and these data may be linked in part with the cognitive impairment observed. Alterations in spine density correspond to aberrant brain function observed in various neurodevelopmental and neuropsychiatric disorders and Density decreases during adolescence, reaching a stable level in adulthood. Previously, it was demonstrated that in the offspring of diabetic rats, the neuronal density in CA3 was reduced, although the neuronal density in CA1 and CA2 showed a decrease but it was not significant (Tehranipour, M. et al 2008. J. Biol. Sci., 8, 1027-1032). On the other hand, it has been demonstrated that developmental programming encompasses the study of a wide range of intrauterine environments in a variety of species and correlates these with diverse phenotypic outcomes in the offspring. Thus, the range of both dependent and independent variables studied often makes the developmental programming complex to interpret and the drawing of definitive conclusions difficult. Therefore, the communal, though little-known, theme of many developmental models is a sex difference in offspring outcomes. One possibility is that in this study rats show no significant differences in total dendritic spine density between the CN and DR groups, indicating that gestational diabetes could have a significant impact on the dendritic spine density which could depend specifically on the gender. Although we did not check if there was an effect on the diverse parameters performed between female and male rats, it would be interesting to analyse if there are sex differences.
Q6. What was the blood glucose level in the rat offspring? Was it recorded during the investigation?
Response: We evaluated the blood glucose in the rat offspring at F21, P0 and P30, as detailed in Table 1.
Q7. Suggestions: should be presented in mmol/L instead of mg/dL for greater clarity. Regarding Tables, the data should be presented as mean ± SEM rather than mean (SEM) for better clarity.
Response: We have modified the blood glucose levels in Figure 1 as suggested. The data in all the tables is now presented as mean ± SEM.
Finally, We want to thank the reviewer for their valuable comments and constructive criticisms. We have taken this into account in this manuscript and for the future.

Round 2
Reviewer 1 Report
Comments and Suggestions for Authors
Table 1. Brain weigh(t)
The authors should respond/address the general/major comment.
Author Response
We are sorry for the typo in Table 1. Brain weight - now is corrected.
In the first version, we thought that the reviewer referred to the fact that the measurement had not been made, without us realizing that there was a typo.
We want to thank the reviewer deeply for pointing out our typo. We have now considered this in this manuscript and the future.
Reviewer 2 Report
Comments and Suggestions for Authors
The authors have addressed the previous concerns regarding the manuscript.
Comments on the Quality of English LanguageMinor editing of English language is required to meet the standard of the journal.
Author Response
Minor editing of English language is required to meet the standard of the journal.
Response: The manuscript has been reviewed by an experienced editor whose first language is English and who specializes in editing papers written by scientists whose native language is not English. However, if it is necessary for a new edition, we will gladly resend it.
Round 3
Reviewer 1 Report
Comments and Suggestions for Authors
The authors should respond to the following comments by providing sentences or a paragraph in the discussion.
The manuscript describes the impact of maternal STZ-induced hyperglycemia on offspring by orthodox neuroscience methods. Accordingly, certain hippocampus-related phenotypes are observed, but it is difficult to attribute these findings solely to hippocampal origins. For instance, motor initiation reluctance due to the weight gain may contribute to the poorer performance of DR mice. The authors then describe morphological changes, specifically decreases in spine density, in the hippocampal CA3. These changes correlate well with the observed phenotypes, but the molecular mechanisms that underlie these changes remain undescribed.
Author Response
The manuscript describes the impact of maternal STZ-induced hyperglycemia on offspring by orthodox neuroscience methods. Accordingly, certain hippocampus-related phenotypes are observed, but it is difficult to attribute these findings solely to hippocampal origins. For instance, motor initiation reluctance due to the weight gain may contribute to the poorer performance of DR mice. The authors then describe morphological changes, specifically decreases in spine density, in the hippocampal CA3. These changes correlate well with the observed phenotypes, but the molecular mechanisms that underlie these changes remain undescribed.
Response: We thank the reviewer very much for this comment. We agree that certain hippocampus-related phenotypes observed are not solely attributed to hippocampal origins. We discussed that other areas are participating, such as the cerebral cortex (pages 10, paragraphs 314-317) and the forebrain and striatum (pages 10-11, paragraphs 332-335). Also, we discussed that maternal hyperglycemia induces weight alterations and neurodevelopment delay consistent with the cognitive impairments observed (page 10 paragraphs 290-299). In relation to some of the molecular mechanisms, several reports have demonstrated that maternal hyperglycemia was accompanied by apoptosis, neuroinflammation in rat offspring hippocampus (page 10 paragraphs 308-310), and oxidative stress (increase in lipid peroxidation, abnormalities in glutathione metabolism, and decreased antioxidant levels) (page 10 paragraphs 314-317). Finally, in the conclusion we indicate that: Further research is necessary to elucidate the underlying mechanisms driving hippocampal alterations caused by gestational diabetes in offspring (pages 14-15 paragraphs 531-542).
Once again, we want to thank the reviewer for their valuable comments and constructive criticisms.
Round 4
Reviewer 1 Report
Comments and Suggestions for Authors
ok